# Structural consequences of turnover-induced homocitrate loss in nitrogenase

Rebeccah A. Warmack ●[1] ✉, Ailiena O. Maggiolo ●[1], Andres Orta ●[2], Belinda B. Wenke[1], James B. Howard[3] & Douglas C. Rees ●[1,4] ✉

Nitrogenase catalyzes the ATP-dependent reduction of dinitrogen to ammonia during the process of biological nitrogen fixation that is essential for sustaining life. The active site FeMo-cofactor contains a [7Fe:1Mo:9S:1C] metal-locluster coordinated with an *R*-homocitrate (HCA) molecule. Here, we establish through single particle cryoEM and chemical analysis of two forms of the *Azotobacter vinelandii* MoFe-protein – a high pH turnover inactivated species and a ΔNifV variant that cannot synthesize HCA – that loss of HCA is coupled to α-subunit domain and FeMo-cofactor disordering, and formation of a histidine coordination site. We further find a population of the ΔNifV variant complexed to an endogenous protein identified through structural and proteomic approaches as the uncharacterized protein NafT. Recognition by endogenous NafT demonstrates the physiological relevance of the HCA-compromised form, perhaps for cofactor insertion or repair. Our results point towards a dynamic active site in which HCA plays a role in enabling nitrogenase catalysis by facilitating activation of the FeMo-cofactor from a relatively stable form to a state capable of reducing dinitrogen under ambient conditions.

Biological nitrogen fixation occurs within diazotrophic organisms primarily via the Mo-nitrogenase[1–4]. This enzyme consists of two air-sensitive metalloproteins, the catalytic molybdenum iron (MoFe-) protein, and the reductase iron (Fe-) protein. The Fe-protein mediates ATP-dependent electron transfer (ET) through its [4Fe:4S] cluster to the intermediate [8Fe:7S] P-cluster in the MoFe-protein; the electrons are subsequently shuttled to a [7Fe:1Mo:9S:1C]-*R*-homocitrate metal-locluster, the FeMo-cofactor, within the active site of the MoFe-protein. The reduction of dinitrogen to ammonia catalyzed by nitrogenase proceeds at ambient temperatures and pressures, and is increasingly of interest as an alternative to the industrial processes for ammonia and hydrogen production, as well as for the reduction of carbon-containing substrates to unsaturated hydrocarbons for biofuels[5,6]. Nitrogenase further serves as a model system for the study of biological metal-catalyzed enzymatic reactions. Despite its significance, the atomic basis for the mechanism is incomplete due to the transient nature of its numerous intermediate states and the challenges of working with these oxygen-sensitive proteins[7].

The FeMo-cofactor remains a primary focus of nitrogenase research efforts, as substrates are expected to bind to and be reduced by this metallocluster. This ligand is attached to the catalytic MoFe-protein through two residues linked to Fe1 and Mo at opposing ends of the cluster; the coordination sphere of Mo is completed by *R*-homo-citrate (HCA) in a bidentate manner. HCA has been proposed to participate in proton transfer to the active site[8] as well as to undergo changes in Mo coordination that could open up sites for substrate binding[9–11]. Structure-activity studies conducted by Ludden and cow-orkers have established that the key features of the HCA ligand for substrate reduction include a hydroxyl group and a minimum of two carboxyl groups[12]. However, the precise structural and chemical role played by HCA in nitrogenase nitrogen fixation remains unclear. Given the possible involvement of HCA in proton transfer, changes in pH

[1]Division of Chemistry and Chemical Engineering 147-75, California Institute of Technology, Pasadena, CA 91125, USA. [2]Biochemistry and Molecular Biophysics Graduate Program, California Institute of Technology, Pasadena, CA 91125, USA. [3]Department of Biochemistry, University of Minnesota, Minneapolis, MN 55455, USA. [4]Howard Hughes Medical Institute, California Institute of Technology, Pasadena, CA, USA. ✉e-mail: rwarmack@caltech.edu; dcrees@caltech.edu

present an avenue for probing the role of this moiety in catalysis. A mechanism-based inactivated state of nitrogenase prepared by turnover under high pH conditions has been described[13]. Yang et al. noted that increased pH drove progressive, partially reversible inhibition of the MoFe-protein in a first-order process, and finally inactivation, with increased inactivation rates observed at higher pHs. The structure of that high pH turnover-inactivated state is still undetermined, as it has proven recalcitrant to crystallization.

In this work, we sought to determine the structure of this turnover-induced state, here referred to as the MoFe[Alkaline-inactivated] state, by employing the cryo-electron microscopy (cryoEM) method of single particle analysis. A resulting 2.37 Å resolution structure of MoFe[Alkaline-inactivated] demonstrates that HCA is lost from the active site under these conditions. We develop a biochemical assay for the detection of HCA from protein samples, and confirm that intact HCA cannot be detected in samples of MoFe[Alkaline-inactivated]. In this structure, the loss of HCA couples to an asymmetric α-subunit domain and FeMo-cofactor disordering, and the formation of a histidine quartet near the more disordered active site. Complementary structural and biochemical analysis of MoFe-protein purified from a homocitrate synthase (ΔnifV) deletion strain of Azotobacter vinelandii, termed the MoFe[ΔNifV] state, reveals loss of HCA and similar structural features. Intriguingly, reconstructions from a subset of particles of this purified protein reveal MoFe[ΔNifV] complexed to an endogenous protein that we have identified through a combination of structural and proteomic approaches, and the structural prediction program AlphaFold, as the previously uncharacterized protein Avin_01560 (also annotated as NafT) present in the nif operon region of the A. vinelandii genome[14,15]. This complex demonstrates the physiological relevance of the homocitrate-compromised form of the MoFe-protein, as NafT specifically recognizes this state. Our results point towards a dynamic active site in which homocitrate plays an active role in enabling the chemistry of nitrogenase catalysis as part of a transformation where the FeMo-cofactor is activated from a relatively stable form to a state capable of reducing dinitrogen under ambient conditions. This work demonstrates that increasingly powerful cryoEM techniques can fundamentally advance our understanding of complex metalloenzyme mechanisms by revealing previously uncharacterized states of relevance to turnover, assembly, or repair.

## Results

### Anaerobic cryoEM structures of MoFe-protein

Single particle cryoEM is ideally suited to interrogate this system given its increasing gains in resolution[16,17] and ability to anaerobically probe macromolecular structures[18–20]. To compare relevant reference states, we pursued cryoEM structures of nitrogenase in solution by preparing an anaerobic workflow for the generation of single-particle cryoEM samples within a glove box. Comparisons of datasets with varying dose, exposure to oxygen, grid types, and conditions were used to benchmark this procedure (Supplementary Figs. 1, 2). Adsorption of proteins to the air–water interface (AWI) has long been associated with apparent discrepancies in expected protein concentration on the grid, particle orientation bias, and protein denaturation[21,22]. Indeed, the determination of intact MoFe-protein structures required either surfactants or the use of carbon-layered grids, due to the localization of MoFe[As-isolated] to the air AWI, causing strong orientation bias, and disorder of its α-subunit (Supplementary Fig. 2)[23–25]. Without the use of surfactants or carbon-layered grids, we observed that the MoFe-protein adheres to the AWI regardless of ice thickness, and displays asymmetric loss of density corresponding to residues in the αIII domain. The use of cryo-electron-tomography (cryoET) was crucial for the validation of the MoFe-protein grid and freezing conditions[23,24]. Optimized vitrification of the as-isolated MoFe-protein alone (MoFe[As-isolated]) yielded a 2.13 Å resolution structure in the presence of detergent (Fig. 1a and Supplementary Fig. 3).

### Structure of the alkaline turnover-inactivated MoFe-protein

Following the determination of the intact MoFe[As-isolated] state, we applied single particle cryoEM to the high pH turnover-inactivated state, designated MoFe[Alkaline-inactivated] (Fig. 1b, c). MoFe-protein was separated from alkaline acetylene reduction reactions, run either with ATP (MoFe[Alkaline-inactivated]) or without ATP (MoFe[Alkaline]), by size exclusion chromatography at pH 7.8 (S.E.C.; Supplementary Fig. 4). The MoFe[Alkaline-inactivated] state displayed a shifted elution time relative to MoFe[Alkaline] and MoFe[As-isolated] indicating a change in the hydrodynamic radius of the protein as previously noted[13]. Electron paramagnetic resonance (EPR) spectroscopy of these states revealed the loss of the canonical as-isolated $S = 3/2$ signal from the FeMo-cofactor in the MoFe[Alkaline-inactivated] form (Supplementary Fig. 4). By cryoEM, the control MoFe[Alkaline] cryoEM structure at 2.14 Å resolution incubated at pH 9.5 closely resembled MoFe[As-isolated] (Fig. 1b and Supplementary Fig. 5). In contrast, the MoFe[Alkaline-inactivated] cryoEM structure exhibited asymmetric disorder within one α-subunit, which likely stems from the averaging of heterogeneous intermediate states. This poor density corresponds to residues 1–48, 354–360, 376–416, and 423–425, which lack traceable density in the more disjointed α-subunit (Fig. 1c and Supplementary Fig. 6; 2.37 Å resolution). These αIII domain residues have been previously shown to rearrange as seen in the FeMo-cofactor-deficient ΔnifB MoFe-protein[26] and more recently in cryoEM structures of the nitrogenase complex[10,18,19]. It should be noted that in our studies, similar changes in the α-subunit were also observed in the cryoEM maps of the AWI-perturbed MoFe[As-isolated] protein. However, the MoFe[Alkaline-inactivated] cryoEM dataset was collected in the presence of detergent to mitigate this problem, and we directly confirmed that the protein was not localized to the AWI as determined by cryoET (Supplementary Fig. 6).

In the MoFe[Alkaline-inactivated] structure, both P-cluster ligands and one FeMo-cofactor appear well ordered, but within the active site of the disordered α-subunit, the density corresponding to the crystallographic location of FeMo-cofactor appears distorted (Fig. 2c). The coordinating Cys α275 loop is near its expected position and appears to remain coordinated to the distorted cofactor density in that region (Fig. 2c). The disordered active site clearly lacks density for HCA, while the ordered active site shows reduced density at the HCA coordination site (Fig. 2c). We developed a method for the quantification of HCA extracted from MoFe-protein samples using ion chromatography coupled with mass spectrometry (IC-MS), revealing that while all MoFe[Alkaline] samples analyzed contained approximately the expected 2 moles HCA per mole MoFe-protein, no HCA was detected in the MoFe[Alkaline-inactivated] samples (Fig. 2d). This loss of homocitrate density in the MoFe[Alkaline-inactivated] maps and the lack of HCA detection by IC-MS may be due to a chemical alteration of the moiety that results in a change of its mass and increased positional disorder within the structure. While the mechanism of this reaction is unknown, we note that aconitase catalyzes the dehydration and rearrangement of the hydroxyacid isocitrate at an iron-sulfur cluster[27]. Inductively-coupled plasma mass spectrometry (ICP-MS) metal analysis of MoFe[Alkaline] and MoFe[Alkaline-inactivated] demonstrated modest, but statistically insignificant, decreases of 2 moles Fe and 0.4 moles Mo per mole MoFe[Alkaline-inactivated] with respect to the control MoFe[Alkaline] (Fig. 2e, f).

### His-coordination site formation within the high pH turnover-inactivated state

Within both αβ dimers of the MoFe[Alkaline-inactivated] state, Phe α300 experiences a rotamer rearrangement with respect to the MoFe[As-isolated] and MoFe[Alkaline] states (Fig. 3). The movement of this bulky side chain within the disordered dimer is coupled with altered positioning of three histidines, His α274, His α362, and His α451, towards a coordination site for an unidentified ligand ~8 Å from the terminal Fe1 of the FeMo-cofactor. In addition, while density around the FeMo-cofactor is poorly resolved, a rearrangement of the His α442 loop that normally ligates the

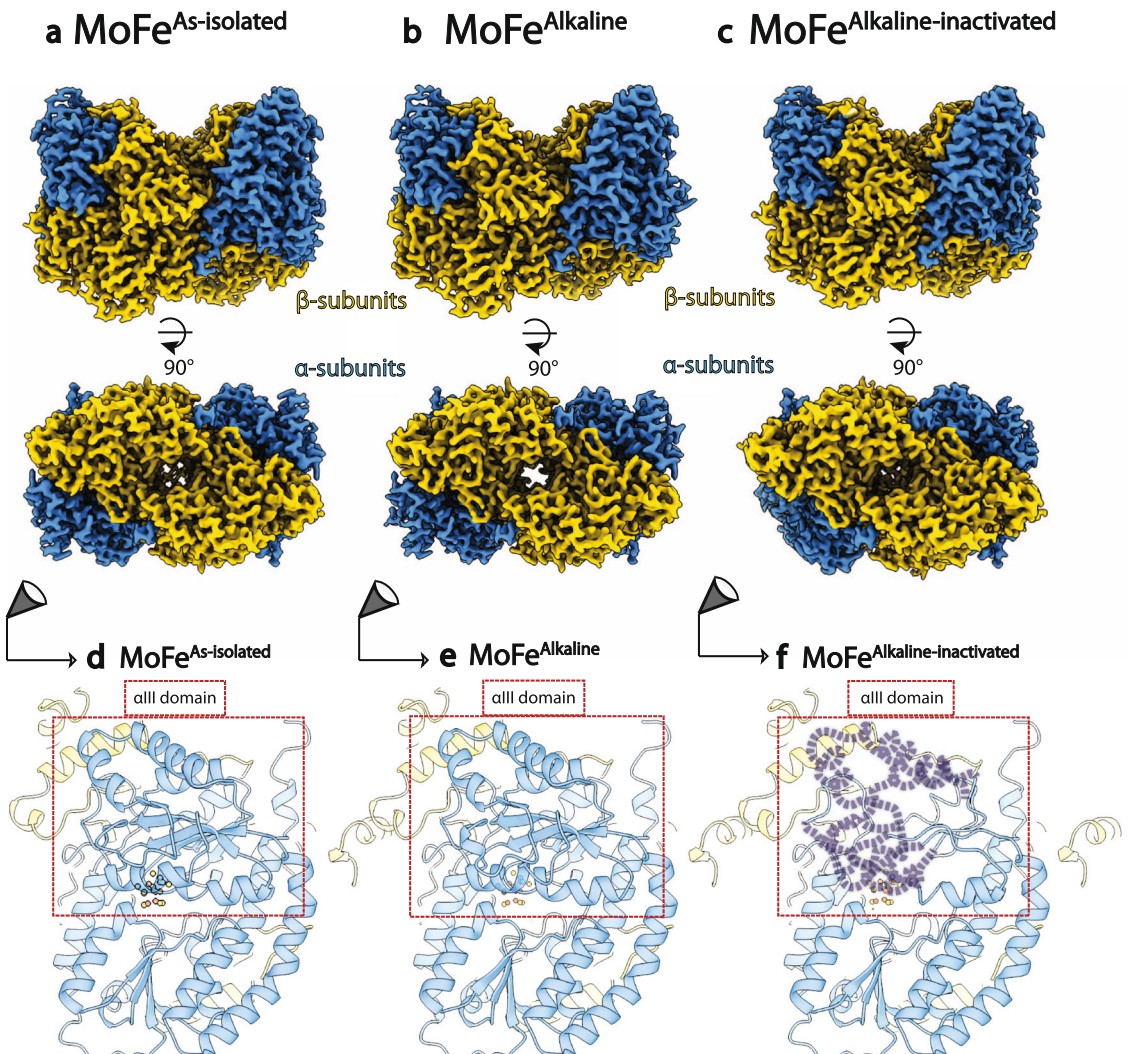

**Fig. 1 | Anaerobic cryoEM structures of MoFe-protein. a** The 2.13 Å resolution cryoEM map of MoFe[As-isolated] purified from *A. vinelandii*. **b** The 2.14 Å resolution cryoEM map of MoFe[Alkaline] from a control acetylene reduction reaction performed at pH 9.5 without ATP isolated via S.E.C. conducted at pH 7.8, demonstrating the same overall architecture as MoFe[As-isolated]. **c** The 2.37 Å resolution cryoEM map of MoFe[Alkaline-inactivated] from an acetylene reduction reaction performed at pH 9.5 with ATP isolated via S.E.C. conducted at pH 7.8, displaying asymmetric disorder within the α-subunit density. **d–f** Location of disorder within the αIII domain of the α-subunit in the **d** MoFe[As-isolated], **e** MoFe[Alkaline], and **f** MoFe[Alkaline-inactivated] structures. α-subunits are illustrated in blue and β-subunits are illustrated in yellow. Structures were solved in the presence of CHAPSO to prevent interactions with the air–water interface.

FeMo-cofactor can be partially traced into smoothed maps and may also participate in this coordination in a tetrahedral fashion. Given the nature of the density and the geometry of the coordination, the bound ligand could be metal. While the conformation of Phe α300 is also altered in the ordered αβ dimer of the MoFe[Alkaline-inactivated] protein, His α362, His α442, and His α451 residues in this ordered subunit remain close to their as-isolated positions, precluding the formation of an equivalent coordination site. Changes in the rotamer state of Phe α300, His α274, and His α451 were recently described under turnover conditions, pointing to the possible significance of these residues to substrate reduction[10]. The additional changes in His α362 and His α442 found in MoFe[Alkaline-inactivated] suggest that the formation of a coordination site is a crucial role for these residue rearrangements and mutations in His α274, His α362, and His α451 correlate to decreased cofactor insertion and activity[28,29]. Thus, the loss of HCA correlates with the disordering of domain αIII within the α-subunit, distortion of the cofactor density, and the formation of a new His-coordination site between α274, His α362, His α451, and His α442.

Analysis of the MoFe[Alkaline-inactivated] structure reveals two further conformational changes in side chains Trp α253 and Gln β93 with respect to MoFe[As-isolated] and MoFe[Alkaline] (Supplementary Fig. 7 and Supplementary Table 2). A distinct Trp α253 rotamer was recently shown to exist under $N_2$ turnover conditions, and has also been previously implicated in the control of substrate access to the active site[10,30]. The same conformational change is observed in the MoFe[Alkaline-inactivated] structure, and may suggest that both acetylene and $N_2$ substrates utilize similar substrate access channels. Interestingly, the Trp α253 flip is observed in both α-subunits in MoFe[Alkaline-inactivated], where previously it was observed only in one subunit under turnover conditions[10]. Additionally, Gln β93 within MoFe[Alkaline-inactivated] experienced a flip away from its resting state position in the β-subunit adjacent to the disordered α-subunit. This residue has been suggested to be along the path of egress for the $NH_3$ product, and is also positioned close to the P-cluster[31].

Three-dimensional variability analysis (3DVA) of the MoFe[Alkaline-inactivated] cryoEM structure revealed the ordering and disordering of the α-subunits (Supplementary Movie 1), indicating the final reconstruction contains a mixture of fully ordered, partially disordered, and fully disordered α-subunits[32]. These changes appear correlated with a breathing motion within the α-subunit. A morph

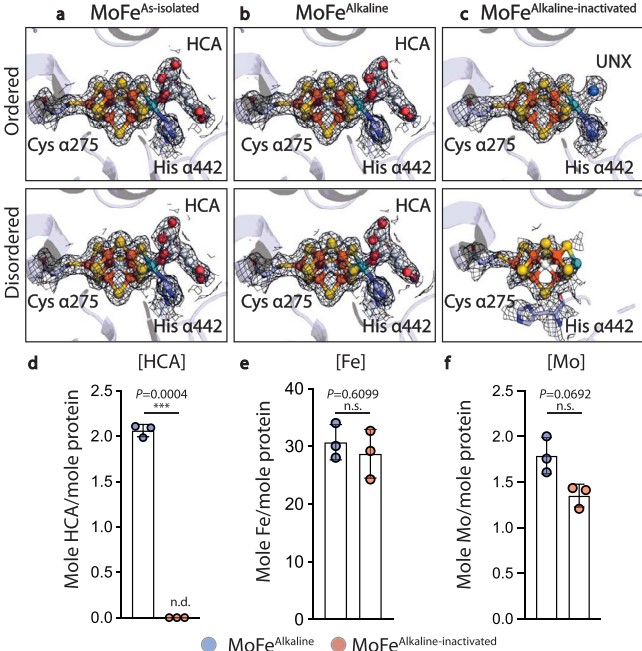

**Fig. 2 | Structure and composition of the alkaline turnover-inactivated MoFe-protein active site. a** CryoEM density carved around the FeMo-cofactor at 7 σ in MoFe[As-isolated] purified from *A. vinelandii* demonstrating the presence of the FeMo-cofactor coordinated by Cys α275, His α442, and an HCA in both αβ dimers. **b** CryoEM density carved around the FeMo-cofactor at 7 σ in MoFe[Alkaline] isolated from a control acetylene reduction reaction mixture at pH 9.5 without ATP, demonstrating the maintenance of the FeMo-cofactor coordinated by Cys α275, His α442, and an HCA in both αβ dimers. **c** CryoEM density carved around the FeMo-cofactor at 7 σ in MoFe[Alkaline-inactivated] isolated from an acetylene reduction assay at pH 9.5 with ATP, demonstrating the presence of the FeMo-cofactor coordinated by Cys α275, His α442, and an unknown ligand (UNX) in the HCA site in the ordered αβ dimer (top panel). The loss of recognizable density for the FeMo-cofactor and the HCA is evident in the disordered αβ dimer (lower panel). **d** Quantification of homocitrate by an IC-MS method developed for this work illustrates the approximately expected 2 moles HCA per mole of tetrameric protein in MoFe[Alkaline] (blue circles), however, HCA was not detected (n.d.) within the MoFe[Alkaline-inactivated] samples (red circles), which have been represented as zero values in this data. **e** ICP-MS metal quantification of Fe illustrates a decrease from the mean of 30.7 moles Fe per mole protein in MoFe[Alkaline] to a mean of 28.7 moles Fe per mole tetrameric protein in MoFe[Alkaline-inactivated]. **f** ICP-MS metal quantification of Mo shows a decrease from the mean of 1.79 moles Mo per mole protein in MoFe[Alkaline] to a mean of 1.35 moles Mo per mole tetrameric protein in MoFe[Alkaline-inactivated]. Each data point in panels **d**–**f** represents a protein isolated from an independent reaction and columns are shown as the mean ± sd. Statistical analyses of three distinct replicates shown in panels **d**–**f** used paired *t*-tests and two-tailed *P* values greater than 0.05 were considered not significant (n.s.). Source data are provided as a Source Data file. HCA *R*-Homocitrate, Fe iron, Mo molybdenum.

between the MoFe[Alkaline-inactivated] and MoFe[As-isolated] structures illustrates a similar movement, with shifts apparent in αIII domain helices Pro α302-Lys α315, Glu α318–Arg α345, and Lys α426–Met α434, and β-strand Lys α349–Ile α355 (Supplementary Movie 2). Intermediate subsets corresponding to the variably ordered states identified in 3DVA were distinguishable across five maps with either two disordered α-subunits or asymmetrically disordered α-subunits (Supplementary Fig. 8). Models built into these intermediate MoFe[Alkaline-inactivated] maps highlight several features. All maps lack HCA density, emphasizing the consistent loss of this ligand. Additionally, as the order increases within one of the α-subunits, cofactor density order is regained, Phe α300 returns to its resting state position, and the His-coordination site is lost.

## The homocitrate-deficient MoFe[ΔNifV] structure resembles the MoFe[Alkaline-inactivated] state

To determine whether the features of the MoFe[Alkaline-inactivated] state stem solely from the loss of HCA, we characterized the homocitrate-deficient MoFe-protein purified from the homocitrate synthase (*nifV*) deletion strain of *A. vinelandii* (MoFe[ΔNifV])[33]. Like MoFe[Alkaline-inactivated], MoFe[ΔNifV] showed similar SEC shifts in elution time relative to MoFe[As-isolated] and had diminished levels of acetylene reduction (Supplementary Fig. 9). Likewise, EPR spectroscopy revealed a loss of the canonical as-isolated $S = 3/2$ signal from the FeMo-cofactor in the MoFe[ΔNifV] form, comparable to the MoFe[Alkaline-inactivated] state. The single particle cryoEM structure of MoFe[ΔNifV] also showed asymmetric disordering of residues 14–19, 25–26, and 408–417 in one of the α-subunits (Fig. 4a and Supplementary Fig. 10). No density was observed at the canonical HCA position within the active site of the disordered α-subunit, but a smaller density remains in the HCA pocket within the corresponding ordered α-subunit (Fig. 4b). No homocitrate was detected by IC-MS, but citrate was isolated from the MoFe[ΔNifV] sample at a molar ratio of 0.2-mole citrate per mole MoFe[ΔNifV]. For comparison, a reduced occupancy of citrate (50%) was previously reported in ΔNifV MoFe-protein isolated from *Klebsiella pneumoniae* (PDB code 1H1L[34]; Fig. 4c). Variations in citrate incorporation between this study and ours may stem from differences between the bacteria or growth media, but in both cases, sub-stoichiometric levels of citrate were observed.

The *K. pneumoniae* ΔnifV crystal structure also lacked the disorder within one α-subunit, perhaps suggesting that its crystal lattice favored more ordered states of the α-subunit. In our work, ~39 moles Fe and 1.8 Mo per molecule of MoFe[ΔNifV] were detected by ICP-MS of the purified protein, suggestive of a full complement of clusters (Fig. 4d). The Phe α300 side chain is flipped in the disordered α-subunit of MoFe[ΔNifV], however, His α362, His α442, and His α451 remain close to their resting state positions in both dimers, preventing the formation of the histidine coordination site. Subsets of these MoFe[ΔNifV] particles isolated by 3DVA did not demonstrate an equivalent coordination site (Supplementary Fig. 8). In addition, while Gln β93 remained close to the as-isolated position in MoFe[ΔNifV], Trp α253 was observed to undergo the same conformational change as observed in the MoFe[Alkaline-inactivated] state in its more ordered α-subunit (Supplementary Fig. 7 and Supplementary Table 2).

## Structure of a previously uncharacterized MoFe[ΔNifV]-NafT complex revealed by cryoEM

Further data collection and classification of a subset of MoFe[ΔNifV] particles resulted in a 2.71 Å overall resolution map with a protruding density that we hypothesized belonged to an endogenous binding partner pulled down with MoFe[ΔNifV] during purification (Fig. 5a and Supplementary Fig. 11). This density appears with more disordered MoFe[ΔNifV] particles (Supplementary Table 2 and Supplementary Fig. 8). A partial model built into this additional density was cross-referenced against Alpha Fold-predicted models of proteins identified in purified MoFe[ΔNifV] samples by proteomic analysis[35–37]. This approach revealed that the AlphaFold model of an uncharacterized protein of 15 kDa, a product of the Avin_01560 gene on the *nif* operon, matched the extended density (Fig. 5a). This gene product has also been annotated as the nitrogenase-associated factor protein T (NafT) in previous studies[14,15]. Building the model into this density yields the MoFe[ΔNifV]-NafT complex (Fig. 5a, b). To distinguish NafT from the MoFe-protein α- and β-chains, we refer to this subunit as the epsilon (ε) chain when designating residues. There is greater relative disorder and lower resolution within the region of the NafT, likely due to residual heterogeneity in the particles. The binding site for this protein on MoFe[ΔNifV] partially overlaps with that of the nitrogenase Fe-protein in the MoFe-protein:Fe-protein ADP-AlF₄⁻ stabilized complex and buries

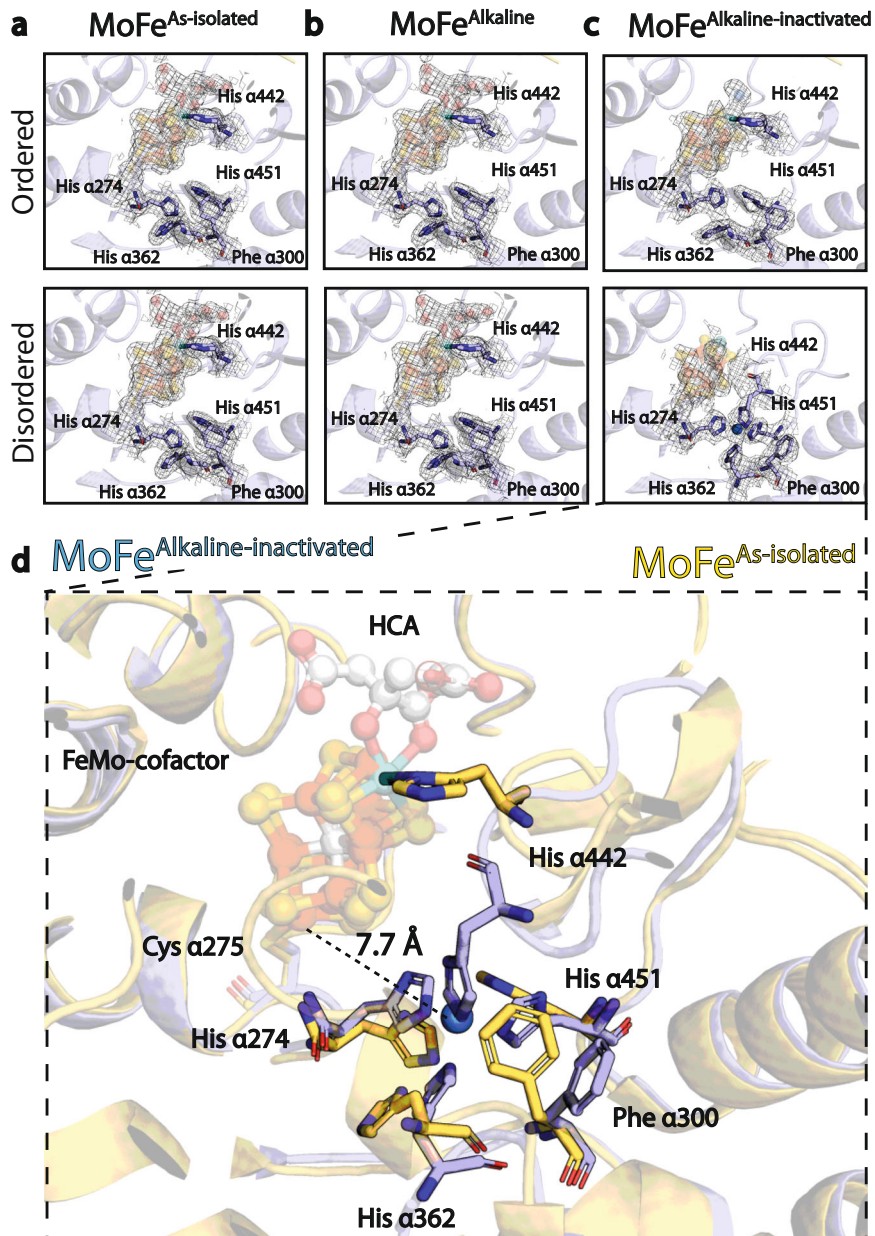

**Fig. 3 | His-coordination site formation within the high pH turnover-inactivated state. a** His α274, His α362, and His α451 within MoFe[As-isolated] purified from *A. vinelandii*. The upper panel displays the His-coordination site in the more ordered dimer, while the lower panel displays the more disordered dimer. **b** His α274, His α362, and His α451 in MoFe[Alkaline] from an acetylene reduction reaction performed at pH 9.5 without ATP isolated via S.E.C. conducted at pH 7.8. The upper panel displays the His-coordination site in the more ordered dimer, while the lower panel displays the more disordered dimer. **c** His α274, His α362, and His α451 in MoFe[Alkaline-inactivated] from an acetylene reduction reaction performed at pH 9.5 with ATP isolated via S.E.C. conducted at pH 7.8. The upper panel displays the His-coordination site in the more ordered dimer, while the lower panel displays the more disordered dimer. The repositioning of the His α442 side chain is evident in the disordered dimer. **d** Enlarged overlay of the active sites and His-coordination site from the disordered dimers of MoFe[As-isolated] (yellow) and MoFe[Alkaline-inactivated] (blue). All densities shown are carved around the atoms at 7 σ. HCA *R*-Homocitrate.

approximately ~1,400 Å² surface area (PDB 1N2C[38]; Fig. 5c, d). At the interface between the two proteins, a well-ordered amphipathic helix from Trp ε117 to Ile ε131 packs against the MoFe[ΔNifV] protein α- and β-subunits. Additionally, a loop in NafT spanning residues Lys ε10 to Leu ε17 extends into the MoFe[ΔNifV] α-subunit. This extension allows for hydrogen bonding between Ser ε14 and Tyr α281, and a salt bridge between Arg ε16 and Asp α200. Perhaps more strikingly, this loop adds two histidines (His ε9 and His ε11) to a cluster of histidines that includes His ε95, His α195, and His α196, four of which are situated within 10 Å of each other (Fig. 5d). It is tempting to speculate that under certain conditions this histidine cluster may be capable of metal or cluster

binding. While the function of the NafT protein has yet to be elucidated, this structure demonstrates that it recognizes homocitrate-deficient forms reminiscent of the MoFe[Alkaline-inactivated] state.

In the MoFe[ΔNifV]-NafT state, the MoFe[ΔNifV] shows asymmetric disordering of residues 1–48, 376–383, 390–398, and 402–409 in one of its α-subunits (Fig. 5). NafT was also disordered from residue 41–48. The Phe α300 side chain is flipped in both α-subunits. In the ordered subunit, His α274, His α362, and His α451 remain close to their resting state positions. By contrast, in the disordered subunit in concert with the movement of Phe α300, His α362, and His α451 have modified positions, appearing to form a His-triad site with His α274. This site

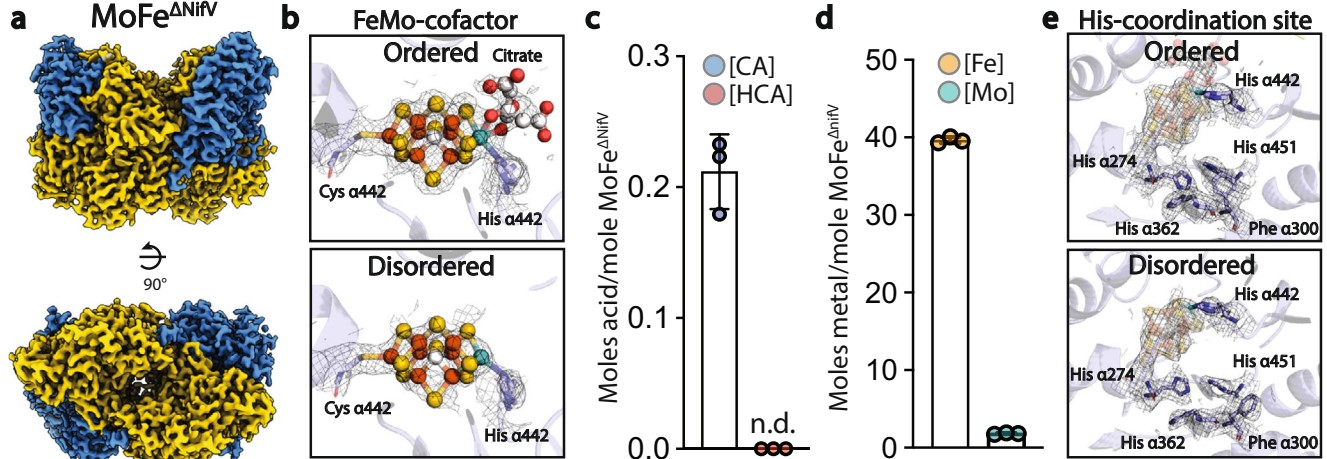

**Fig. 4 | The homocitrate-deficient MoFe$^{\Delta NifV}$ resembles the MoFe$^{Alkaline-inactivated}$ state, with partial occupancy citrate replacing homocitrate in the active site. a** 2.58 Å resolution cryoEM map of MoFe$^{\Delta NifV}$ purified from *A. vinelandii* lacking a functional *nifV* gene product. **b** CryoEM density carved around the FeMo-cofactor at 7 σ in MoFe$^{\Delta NifV}$ demonstrating the presence of the FeMo-cofactor coordinated by Cys α275, His α442, and citrate in place of the HCA site in the ordered αβ dimer (top panel). A considerable loss of recognizable density for the FeMo-cofactor and the HCA is evident in the disordered αβ dimer (bottom panel). **c** Quantification by an IC-MS method developed for this work confirms the presence of citrate (CA; blue circles) within the purified MoFe$^{\Delta NifV}$ protein at very low levels (0.2 moles citrate per mole protein). HCA was not detected (n.d.) in this sample (red circles). **d** ICP-MS metal quantification shows ~39 moles iron (Fe; orange circles) and ~1.8 moles molybdenum (Mo; teal circles) per mole purified MoFe$^{\Delta NifV}$ protein, respectively. **e** CryoEM density carved at 7 σ around His α274, His α362, and His α451 in MoFe$^{\Delta NifV}$ equivalent to the coordination site observed in MoFe$^{Alkaline-inactivated}$. In panels **c**, **d**, *n* = 3 independent reads of the same sample and columns are shown as the mean ± s.d. Source data are provided as a Source Data file. HCA *R*-Homocitrate, CA citrate, Fe iron, Mo molybdenum.

differs from that in the MoFe$^{Alkaline-inactivated}$ state in the absence of His α442 (Supplementary Table 2). In both α-subunits, Trp α253 has the same altered conformation as that observed in MoFe$^{Alkaline-inactivated}$, but Gln β93 remains in the as-isolated state (Supplementary Fig. 7).

## Discussion

The features observed in this study, including HCA loss, distorted cofactor density, α-subunit disordering, the formation of a His-coordination site within the MoFe$^{Alkaline-inactivated}$ protein, and the recognition of the similar MoFe$^{\Delta NifV}$ state by the endogenous binding partner NafT, have important implications for the mechanism of nitrogenase. Based upon these observations, summarized in Supplementary Table 2, we suggest that the bidentate coordination of the HCA acts in part as a staple that helps maintain the FeMo-cofactor within the binding pocket. Perturbations in the cofactor site associated with HCA loss are correlated with disordering in the α-subunit as seen in the MoFe$^{Alkaline-inactivated}$ and MoFe$^{\Delta NifV}$ structures. The asymmetry observed in both the MoFe$^{Alkaline-inactivated}$ and MoFe$^{\Delta NifV}$ structures may suggest crosstalk between the two dimers[10,39,40], though the 3DVA results of the MoFe$^{Alkaline-inactivated}$ structure also indicate that both dimers can be either ordered or disordered, as well. Our cryoEM structures further support the possibility of altered coordination or even complete dissociation of HCA and His α442 from Mo during turnover, enabling repositioning of the FeMo-cofactor while it remains tethered to the protein through Cys α275. This accounts for the disordered density observed in the active sites of our MoFe$^{Alkaline-inactivated}$ and MoFe$^{\Delta NifV}$ cryoEM maps. A potential driving force for this transformation could be the preference for hydroxytricarboxylic acid species such as HCA to form tridentate metal complexes[41], with the binding of a second HCA carboxyl group to Mo coupled to the displacement of the His α442 ligand. One consequence of a tridentate coordination mode would be to enable rotation of the cofactor about the Fe1–Cys α275 bond, thereby providing a rationalization for the apparent interconversion of belt sulfur positions under turnover[42,43]. Rotation of the cofactor would avoid the high energetic barrier to internal scrambling noted by Dance[44]. Further, the His-coordination site observed within the turnover-inactivated

MoFe$^{Alkaline-inactivated}$ structure may serve to usher the mobile cluster towards the surface of the protein, either for the expulsion of a damaged state or for catalysis at the surface of the MoFe-protein (Fig. 5d).

While the inactivation reaction and subsequent features of MoFe$^{Alkaline-inactivated}$ described take place at high pH, these processes are also expected to occur at physiological pH albeit at a decreased rate[13]. During slow diazotrophic growth, inactivated MoFe-protein should accumulate, thus highlighting the possible need for a repair mechanism. In vivo disordering, rearrangements observed in the active site-adjacent histidines, and cofactor mobility may recruit binding partners such as NafT and could constitute a mechanism for recognizing damaged states. In other cellular redox systems, such as photosystem II, auxiliary proteins can facilitate the formation of multisubunit complexes or the insertion of metallocofactors. Thus, it is possible that NafT similarly serves as an assembly factor for the recruitment of additional proteins for stabilization, repair, degradation, or catalysis. Given the similarities between the MoFe$^{\Delta NifV}$ and MoFe$^{\Delta NifB}$ states, it is also possible that this protein plays a role in the biogenesis of nitrogenase or cofactor insertion. Further studies are being conducted to understand the relationship between the features observed in the captured MoFe$^{Alkaline-inactivated}$ state, the fate of homocitrate, and the role of NafT in nitrogenase function.

## Methods

### Purification and characterization of MoFe$^{As-isolated}$ and MoFe$^{\Delta NifV}$

The MoFe$^{As-isolated}$ and the Fe-protein$^{As-isolated}$ were purified from wild-type *A. vinelandii* Lipman (ATCC 13705, strain designation OP). MoFe$^{\Delta NifV}$ was purified from a *nifV* deletion strain, DJ605, which was a kind gift from Dr. Dennis Dean[33]. Purifications were performed under anaerobic conditions using a combination of Schlenk line techniques and anaerobic chambers with oxygen-scrubbed argon[13]. Protein concentrations were determined by amino acid analysis at the UC Davis Molecular Structure Facility. Concentrations in mg/mL were calculated based on molecular weights of 230 and 64 kDa for the MoFe-protein and Fe-protein, respectively. The purity of samples were assessed by bottom-up

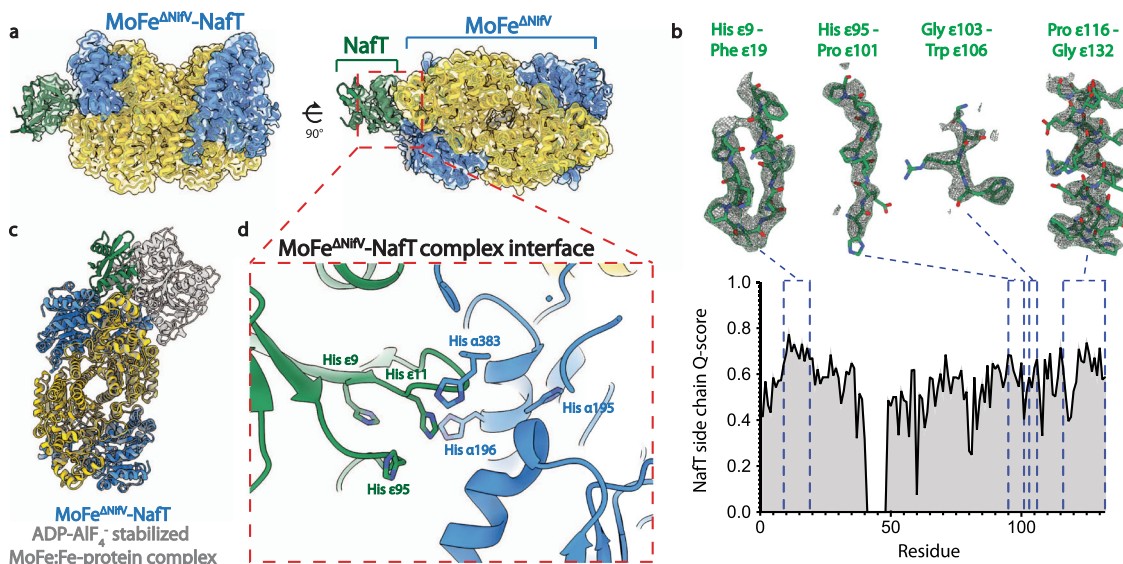

**Fig. 5 | Structure of a previously uncharacterized MoFe$^{\Delta NifV}$-NafT complex revealed by cryoEM. a** Classification of a subset of particles from the MoFe$^{\Delta NifV}$ sample yields a 2.71 Å resolution cryoEM map of MoFe$^{\Delta NifV}$ in complex with the previously uncharacterized protein NafT (chain ε; MoFe$^{\Delta NifV}$-NafT). The atomic model of the MoFe$^{\Delta NifV}$-NafT complex is shown fit into the map. Domains are color-coded as follows: MoFe$^{\Delta NifV}$ α-subunit, blue; MoFe$^{\Delta NifV}$ β-subunit, yellow; NafT, green. **b** Upper panel, density and model corresponding to the designated residues within NafT (chain ε). Lower panel, side chain Q-score values for NafT residues (residues 41–48 were not modeled). **c** Overlay of the MoFe$^{\Delta NifV}$-NafT complex with the structure of the MoFe-protein:Fe-protein ADP-AlF$_4^-$ stabilized complex (PDB 1N2C[38]). **d** Interface between NafT (green) and the MoFe$^{\Delta NifV}$ α-subunit (blue). Source data are provided as a Source Data file. ADP adenosine diphosphate, AlF$_4^-$ aluminum fluoride.

mass spectrometry, as the raw abundance of NifD and NifK was divided by total raw abundance values. Due to the impurity of the MoFe$^{\Delta NifV}$ sample, the concentrations obtained from amino acid analysis were multiplied by the percent purity of NifDK in the solution (30%). The component ratio (CR) of Fe-protein to MoFe-protein was defined as moles of Fe-protein per mole of MoFe-protein active site via the equation CR = 1.82($C_{Fe}$/$C_{MoFe}$), where $C_{Fe}$ and $C_{MoFe}$ are the concentrations of the Fe-protein and MoFe-protein in the final reaction mixture, respectively, in mg/mL. Nitrogenase activity was determined by the reduction of acetylene to ethylene as measured in a 1 mL reaction with 9 mL headspace incubated at 30 °C for 10 min in an argon atmosphere with 1 mL acetylene gas. MoFe-protein and Fe-protein were incubated at varying CRs in reactions with 20 mM sodium dithionite, 5 mM MgCl$_2$, 5 mM ATP, 20 mM creatine phosphate, 23 U/mL creatine phosphokinase, buffered in 50 mM Tris-HCl (pH 7.8) with 200 mM NaCl. Reactions were quenched with 0.25 mL 1 M citric acid and ethylene was quantified by gas chromatography from 40 µL aliquots removed from the headspace. The specific activity for acetylene reduction was ~2300 nmol/min/mg MoFe$^{As-isolated}$ and ~600 nmol/min/mg for MoFe$^{\Delta NifV}$.

### Preparation of high pH turnover-inactivated nitrogenase, MoFe$^{Alkaline-inactivated}$

High pH turnover-inactivation reactions were performed as follows: an ATP regeneration system of 5 mM ATP, 5 mM MgCl$_2$, and 20 mM creatine phosphate was prepared in a tribuffer system composed of 100 mM *N*-(2-acetamido)-2-aminoethanesulfonic acid (ACES), 50 mM tris(hydroxymethyl)aminomethane (Tris), and 50 mM ethanolamine, with p$K_a$ values of 6.67, 8.0, and 9.5 at 30 °C, respectively[13]. The pH of the ATP regeneration system was brought up to pH 9.5 with NaOH. Twenty-three U/mL phosphocreatine kinase were added, and aliquots of the solution were added to anaerobic flasks. Reactions were started with the addition of 0.125 mg/mL MoFe-protein and 0.138 mg/mL Fe-protein final concentration (CR = 2). At 240 min, reactions were concentrated in an Amicon 100 MWCO spin filter and products were separated by size exclusion chromatography. The peak corresponding

to the inactivated MoFe-protein was collected anaerobically and concentrated again in an Amicon 100 MWCO spin filter prior to subsequent analyses.

### Continuous-wave electron paramagnetic resonance spectroscopy

One part protein sample was mixed with one part 50 mM Tris-HCl (pH 7.8), 200 mM NaCl, 5 mM sodium dithionite, and two parts 100% ethylene glycol. Two hundred µL total sample was placed at the bottom of a quartz EPR tube. Data were obtained at the Caltech EPR Facility on a Bruker EMX spectrometer using Bruker Win-EPR software for data acquisition.

### Ion chromatography mass spectrometry quantification of tricarboxylic acids

Homocitrate and citrate from MoFe-protein samples were quantified using a Dionex Integrion HPIC ion chromatography system coupled to an ISQ-EC Single Quadrupole Mass Spectrometer (Thermo Fisher Scientific, San Jose, CA). Chromeleon 7.2.10 software was used for data acquisition. Samples were diluted into 200 µL of 150 µM EDTA in water and subsequently precipitated in acetonitrile at 4 °C overnight. Samples were dried to completion in a Savant Speed Vac, and resuspended in 200 µL MilliQ water. Samples were injected using an AS-AP autosampler operated in push-partial mode, loading 5 µL into a 25 µL sample loop. Components were separated on a Dionex IonPac AS11-HC IC column (product #052960) from 20–50 mM potassium hydroxide over 17 min. A Dionex ADRS 600 suppressor (Thermo Fisher Scientific, San Jose, CA) was used in external water mode to remove the potassium hydroxide before conductivity detection followed by negative ion mode mass spectrometry. Homocitrate and citrate standards were prepared in water over concentration ranges spanning those of the protein. Ion signals were recorded using component mode (selected ion monitoring) at 191 m/z for citrate and 205 m/z for homocitrate.

### Inductively-coupled plasma mass spectrometry metal analysis

Iron and molybdenum contents of MoFe- and Fe-protein samples were quantified using an Agilent 8800 Triple Quadrupole ICP-MS in both

oxygen mode (mass shift) and helium mode (kinetic energy discrimination) running Agilent MassHunter software. Operating parameters were determined following standard protocols recommended by the manufacturer that determined flow rates of 0.4 and 4.2 mL/min for oxygen and helium, respectively. In oxygen mode, iron and molybdenum were determined using 56 to 72 m/z and 95 to 127 m/z, respectively. In helium mode both iron and molybdenum were determined in MSMS mode, 56 to 56 and 95 to 95, respectively.

Standard mixtures containing iron and molybdenum (Periodic table mix 1 for ICP, product #92091; molybdenum standard for ICP-MS, product #04488) were prepared in 5% nitric acid spanning 0–100 µg/L for Periodic table mix 1 and 0–50 µg/L for the molybdenum standard. Ten to 30 µL of protein sample ranging from 3 to 18 mg/mL was diluted into 285 µL of 70% nitric acid and incubated at 50 °C for 3 h to release metals bound to the protein. Samples were then diluted into a final volume of 10 mL MilliQ water. The iron and molybdenum concentrations were determined using $^{56}Fe$ and $^{95}Mo$ isotopes and normalized to the MoFe-protein concentration in the sample.

### Bottom-up mass spectrometry of MoFe$^{\Delta NifV}$ sample
Protein samples were incubated with 7.5 M urea for 15 min at 37 °C, then reduced with 4 mM TCEP for 20 min at 37 °C. Chloroacetamide was added to a final concentration of 12 mM and the samples were incubated for 15 min at 37 °C. About 1.5 ng/µL endoproteinase LysC was added and the digestion proceeded for 1 h at 37 °C. Samples were diluted to 2 M urea with 50 mM HEPES, pH 8.0 was added to dilute to 2 M urea, and 1 mM CaCl₂ final concentration was added. Trypsin was added at a concentration of 0.5 ng/µL and incubated overnight at 37 °C. Samples were desalted using Pierce™ C18 Spin Tips & Columns (Thermo Fisher Scientific, product #89870). After desalting and drying, peptides were suspended in water containing 0.2% formic acid and 2% acetonitrile for further LC-MS/MS analysis. LC-MS/MS analysis was performed with an EASY-nLC 1200 (Thermo Fisher Scientific, San Jose, CA) coupled to a Q Exactive HF hybrid quadrupole-Orbitrap mass spectrometer (Thermo Fisher Scientific, San Jose, CA). Peptides were separated on an Aurora UHPLC Column (25 cm × 75 µm, 1.6 µm C18, AUR2-25075C18A, Ion Opticks) with a flow rate of 0.35 µL/min for a total duration of 75 min and ionized at 1.6 kV in the positive ion mode. The gradient was composed of 6% solvent B (2 min), 6–25% B (20.5 min), 25–40% B (7.5 min), and 40–98% B (13 min); solvent A: 2% acetonitrile (ACN) and 0.2% formic acid (FA) in water; solvent B: 80% ACN and 0.2% FA. MS1 scans were acquired at the resolution of 60,000 from 375 to 1500 m/z, AGC target 3e6, and a maximum injection time of 15 ms. The 12 most abundant ions in MS2 scans were acquired at a resolution of 30,000, AGC target 1e5, maximum injection time 60 ms, and normalized collision energy of 28. Dynamic exclusion was set to 30 s and ions with charges +1, +7, +8, and > +8 were excluded. The temperature of the ion transfer tube was 275 °C and the S-lens RF level was set to 60. MS2 fragmentation spectra were searched with Proteome Discoverer SEQUEST (version 2.5, Thermo Scientific) against in silico tryptic digested Uniprot all-reviewed *A. vinelandii* database. The maximum missed cleavages was set to 2. Dynamic modifications were set to oxidation on methionine (M, +15.995 Da), deamidation on asparagine and glutamine (N and Q, +0.984 Da), and protein N-terminal acetylation (+42.011 Da). The maximum parental mass error was set to 10 ppm, and the MS2 mass tolerance was set to 0.03 Da. The false discovery threshold was set strictly to 0.01 using the Percolator Node validated by *q* value. The relative abundance of parental peptides was calculated by integration of the area under the curve of the MS1 peaks using the Minora LFQ node. The resulting proteomic hits are provided in Supplementary Data 1.

### CryoEM sample preparation and data collection
Dilutions of protein samples were prepared in 50 mM Tris-HCl, 200 mM NaCl, and 5 mM sodium dithionite (pH 7.8) in an anaerobic chamber. Three to four µL of the sample were applied to freshly glow-

discharged Quantifoil R1.2/1.3 300 mesh ultrathin carbon grids and blotted for 1–3 s with a blot force of 6, at ~90% humidity using a Vitrobot Mark IV (FEI) in an anaerobic chamber[18]. In detergent-containing conditions, equivalent volumes of a 0.2–1% (*w/v*) CHAPSO stock were mixed with the protein solution immediately before application to the grid. For aerobic preparation of MoFe$^{Oxidized}$, the protein solution was prepared in 50 mM Tris-HCl, 200 mM NaCl, and 5 mM sodium dithionite (pH 7.8) in an anaerobic chamber. This solution was transferred to a 10 mL Wheaton vial and capped with a rubber septum. The Wheaton vial was removed from the anaerobic chamber. Three to four µL of protein solution was removed from the Wheaton vial using a Hamilton gas-tight syringe equilibrated with an anaerobic solution of 5 mM sodium dithionite in 50 mM Tris-HCl, 200 mM NaCl (pH 7.8). The protein solution was immediately transferred onto a grid within an aerobic benchtop Vitrobot Mark IV (FEI) and plunge-frozen using the blotting conditions described above. Grids were stored in liquid nitrogen until data collection. Datasets were collected with a 6k × 4k Gatan K3 direct electron detector and Gatan energy filter on a 300 keV Titan Krios in superresolution mode using SerialEM[45] at a pixel spacing of 0.325 Å. A total dose of 60 e⁻/Å² was utilized with a defocus range of −0.8 to −3.0 uM at the Caltech CryoEM Facility.

### CryoEM image processing
Processing of datasets was primarily performed in cryoSPARC 3, while masked 3D classification was performed in RELION 4.0[46,47]. Movie frames were aligned and summed using patch motion correction, and the contrast transfer function (CTF) was estimated using the patch CTF estimation job in cryoSPARC. Templates for automated picking in cryoSPARC were generated by manually picking from a subset of micrographs. Picked particles were used for multiple rounds of reference-free 2D classification, ab initio model generation, and heterogeneous refinement (Supplementary Figs. 5, 6, 10, 11). Global CTF refinement was carried out, followed by non-uniform refinement[46]. Values for B-factor sharpening were determined by the Mrc2Mtz program and applied to unsharpened maps in cryoSPARC[48]. Resolutions were estimated by the gold-standard Fourier shell correlation (FSC) curve with a cut-off value of 0.143. Local resolution was calculated using the local resolution estimation job in cryoSPARC. Three-dimensional variability analysis[32] of MoFe$^{Alkaline-inactivated}$ was carried out on 2x downsampled particles following Non-Uniform Refinement in cryosparc 3.3 (156,311 particles; 2.37 Å resolution) with three modes over 20 iterations using a filter resolution of 5 Å. Maps representing variability components were calculated using intermediate mode with five frames and no filters. These maps were re-extracted with no binning and put through homogeneous refinement.

### Model building and refinement
Initial model fitting was carried out in ChimeraX[49] using PDB 3U7Q for the MoFe-protein alone maps and the C1DH13 AlphaFold model for MoFe$^{\Delta NifV}$·NafT[36,37]. For the MoFe$^{Alkaline-inactivated}$ structure, the cryoEM maps were smoothed by the application of a B-factor of 20 Å² for initial model building. This effectively lowers the resolution, thereby reducing noise correlated with higher-resolution data and highlighting the main chain density. Final models were refined into maps sharpened by a B-factor of −50 Å². Multiple iterations were carried out of the manual model building and ligand fitting in Coot[50], and refinement in Refmac5[51] and Phenix[52]. Data collection, refinement, and validation are presented in Supplementary Table 1.

### Cryo-electron tomography data collection and processing
Grids of purified proteins were prepared as described above and imaged in a Talos Arctica electron microscope operating at 200 keV with a 6k × 4k Gatan K3 direct electron detector. Datasets were collected at a pixel size of 1.47 Å using a dose-symmetric acquisition scheme from +63° to −63° within SerialEM to a cumulative dose of

120 e$^-$/Å$^2$ per tilt series. Frames were aligned, motion- and CTF-estimated in Warp[53]. Tomograms from resulting image stacks were reconstructed in IMOD[54] using weighted back projection after patch track alignment. The missing wedge was corrected in IsoNet[55]. Particles were visualized in these 3D volumes in ChimeraX.

### Figure preparation and presentation

Structural figures were prepared in ChimeraX and Pymol[56]. Graphs were generated and statistical analysis were performed using GraphPad Prism version 9.0.0 (GraphPad Software, San Diego, California USA, www.graphpad.com). Figures were compiled in Adobe Illustrator.

### Reporting summary

Further information on research design is available in the Nature Portfolio Reporting Summary linked to this article.

## Data availability

The single particle cryoEM maps and models generated in this study have been deposited into the PDB and EMDB for release upon publication. Reconstructed maps and refined models have been deposited with the following PDB and EMDB codes: 8CRS, EMD-26957 (MoFe$^{As\text{-}isolated}$); PDB 8DBX, EMD-27316 (MoFe$^{Oxidized}$); PDB 8ENL, EMD-28272 (MoFe$^{Alkaline\text{-}inactivated}$); PDB 8ENM, EMD-28273 (MoFe$^{Alkaline}$); PDB 8ENN, EMD-28274 (MoFe$^{\Delta NifV}$); PDB 8ENO, EMD-28275 (MoFe$^{\Delta NifV}$-NafT). The Uniprot all-reviewed *A. vinelandii* database was used for the analysis of mass spectrometry data. All other data are available from the corresponding authors upon reasonable request. Source data are provided with this paper.

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

## Acknowledgements

This work was funded by support from the Howard Hughes Medical Institute (D.C.R.), NIH grant GM045162 (D.C.R.), and NIH GM143836-01 (R.A.W.). The foundational contributions of Dr. Thomas Spatzal to establishing the anaerobic Vitrobot system are gratefully acknowledged. We thank Dr. Jens Kaiser, Dr. Songye Chen, Dr. Trixia Buscagan, and Przemyslaw Dutka for their invaluable discussions. IC-MS and ICP-MS were performed on instrumentation made available by the Resnick Sustainability Institute's Water and Environment Lab at the California Institute of Technology. EPR data were collected at the Caltech EPR Facility which is supported by NSF-1531940. Bottom-up mass spectrometry of protein samples was performed at the Beckman Proteome Exploration Laboratory supported by the Arnold and Mabel Beckman Foundation. We thank Dr. Nathan Dalleska, Dr. Paul Oyala, and Dr. Ting-Yu Wang for their support and assistance with these analyses. The generous support of the Beckman Institute for the Caltech CryoEM Resource Center was essential for the performance of this research. We dedicate this work to our co-author Dr. James B. Howard who passed during the course of this work.

## Author contributions

R.A.W. and D.C.R. designed experiments. R.A.W. performed sample preparation, biochemical analyses, electron microscopy data collection and analyses, model building, and refinement. A.O.M. performed cryoEM model building and refinement. A.O. purified protein from the mutant *A. vinelandii*. B.B.W. performed initial tests of anaerobic cryoEM workflow. J.B.H. provided valuable advice on the project and the manuscript. D.C.R. supervised all research. R.A.W. and D.C.R. wrote the manuscript and all authors contributed to revisions.

## Competing interests

The authors declare no competing interests.
