## [Peer Review File · Nature Communications]

Structural consequences of turnover-induced homocitrate loss in nitrogenaseREVIEWER COMMENTS

Reviewer #1 (Remarks to the Author):

Key results

What are the noteworthy results?

In addition to the metals of the MoFe cluster of nitrogenase, the cluster has a homocitrate (HCA) molecule. The biological rationale behind the presence of HCA in the MoFe cluster has been enigmatic. Previous attempts to crystallize nitrogenase in the absence of HCA were unsuccessful. Here Rees and coworkers have been able to obtain structures of nitrogenase in the absence of HCA at atomic resolution using cryo-electron microscopy. These structural data have been long awaited. Additionally, authors have obtained a structure of a complex between HCA-deficient nitrogenase and a previously uncharacterized protein NafT.

Significance

Will the work be of significance to the field and related fields? How does it compare to the established literature? If the work is not original, please provide relevant references.

Yes, the work will be significant to the field and yes, it is original. Multiple novel structures of nitrogenase are presented. Nitrogenase is an extremely important enzyme as it has the potential of providing a sustainable alternative to the energy-consuming Haber-Bosch process of nitrogen fixation. The scientific community will be very interested in this work.

Data and Methodology

Does the work support the conclusions and claims, or is additional evidence needed?

Yes, the work does support the conclusions and claims – additional evidence I suggested is under the suggested improvements section.

Is the methodology sound? Does the work meet the expected standards in your field?

Yes, the methodology is sound. CryoEM is a quickly evolving field and Rees and coworkers have utilized the most recent tools to solving single particle EM.

Is there enough detail provided in the methods for the work to be reproduced?

Yes, there is enough detail provided to reproduce the results.

Validity

Are there any flaws in the data analysis, interpretation and conclusions? - Do these prohibit publication or require revision?

I do not believe there are flaws in the data analysis, interpretation, and conclusions.

Clarity and context/ References

The text is clear and the references the literature appropriately. There are two sections I would like a few more references and they are noted in the "suggested improvements" section.

Suggested improvements

1. P 5 line 112 "Intentionally blurred maps" are mentioned in the text and not explained further in methods
2. Page 6 Line 119 has a typo: "The additional the significance"
3. Page 9 Line 204 and Extended figure 3 and 8. In some structures shown in Ext Fig 8, it looks like there is density for a metal in the histidine coordination site and there is an orange sphere shown, but authors do not seem to comment on the orange sphere. Has a metal ion been refined in any of the structures? Is the "mononuclear metal site" in Ext Fig 3, the same thing as the "histidine coordination site." Clarity on these points would be helpful.
4. Page 9 Lines 190-191 I don't think that it is clear that the experimental cryoEM density was good enough to validate that NafT is the protein bound to the MoFe protein. If true, it would be good to say that you were able to fit the NafT sequence into the map and validate the AlphaFold prediction. It would be good to show density for sidechains of NafT and the sequence of NafT as an extended figure.
5. Page 9 Lines 205-206 "While the function of the NafT protein has yet to be elucidated, it is clear that it recognizes homocitrate-deficient forms reminiscent of the MoFeAlkaline-inactivated state." I am assuming authors mean that the presence of NafT bound in the EM structure is what makes the recognition clear, but authors might want to rephrase "... it is clear from the structural data presented here..."

Reviewer #2 (Remarks to the Author):

I was asked to review the manuscript because of my expertise in cryo-electron microscopy and not because I am familiar with nitrogenase. The work is exemplary in terms of cryoEM analysis. The resolution is better than average and the models do not show any obvious errors. Also, the fact that CET was done to investigate AWI gives the work a certain charm.

There are only two things I would like to see. First, AWI is a normal problem in the field, and not just specific to nitrogenase. Here I would like to see a bit more discussion. This is something almost every cryoEM study has to take into account.

Personally, I find the small protein NafT the most intriguing. Is this an assembly intermediate or a way to eliminate inactive complexes - dead end assembly? This should be speculated a little more. Especially in the context of other assembly intermediates of other large machines like complex I or PSII. This would place the nitrogenase here in a larger context of the biogenesis of molecular machines.

Reviewer #3 (Remarks to the Author):

I find this paper to be interesting and scholarly. It is clearly written and to my view the experiments are carefully carried out and reported. The mechanism of nitrogenase has been challenging to obtain given all the E-states and difficulties synchronizing the transitions between, but progress has been made by a variety of groups, including of course the Rees laboratory and associates. Thanks to Rees and others we have had a resting state structure of the MoFeCo cluster and the P-cluster and the surrounding protein for ~3 decades now. Some of the popular mechanistic models basically are tuned around a rather static structure with little changing through the states (eg the Hoffman et al Janus/hydride elimination model). But now cryoEM is showing interesting changes (this manuscript, and the recent Tezcan et al Science paper (ref 11 here)). The changes of homocitrate binding observed here, along with a number of parallel observations, are quite interesting. Though this is not a final story as of yet, it points to a more dynamic nitrogenase enzyme structure which should lead to a better idea of how this crucial enzyme actually works. So I am very supportive of publication.

Reviewer #4 (Remarks to the Author):

Warmack et al. aim to describe the role of homocitrate molecule in the catalytic cycle of reducing dinitrogen.

Figure 2 they indicate that in panel e they indicate a drop in Fe from 33 to 26 moles per mole of protein. However, the mean in the graph does not reflect these values. From the graph displayed the values appear to be 30.5 and maybe 28 in the alkaline-inactivated version. The abstract and figure 2. Indicate a molar ratio of 7 Fe and 1 Mo per mole of protein in the dinitrogenase and a 8Fe in the P-cluster thus a total of 15. They measure ~30. How do they reconcile the discrepancy? Appears they are using the "dimer" complex molecular weight. This is ok but the axis needs to be updated for accuracy.

Similarly in (2f) the values in the text don't match the graph. The text indicates mean of 2 and 1.2 but interpolation from the graph yields 1.8 and 1.4 are closer to what is graphed.

For both 2f and 2e they fail to provide any statistical test. Based on the variance in Fe I would suggest that the decrease fails to be significant. They would likely need more replicates to have the power to detect the level of Fe decrease shown. Similar comment for Mo. There was no discussion what the potential decrease meant.

The legend of Figure 2 is missing a description of what color each ball represents.

For the ICP-MS methods they indicate a volume of sample used but not the concentration. Thus it is not possible to deduce the standard curve concentration range used. The information on the standard curve used should be included. Were internal standards used and which ones.

More detailed methods for the ICP-MS are required to reproduce this study. What flow rate of oxygen was used? Also why did they not conduct a mass shift assay if they are using oxygen. Particularly for Mo as it readily forms the double oxide. With a relatively pure sample it seems that using He KED would be appropriate too.

Point by point response to reviewer comments:

We appreciate the constructive and supportive comments of the reviewers and in response have revised the manuscript as detailed below.

Reviewer #1 (Remarks to the Author):

Key results

What are the noteworthy results?

In addition to the metals of the MoFe cluster of nitrogenase, the cluster has a homocitrate (HCA) molecule. The biological rationale behind the presence of HCA in the MoFe cluster has been enigmatic. Previous attempts to crystallize nitrogenase in the absence of HCA were unsuccessful. Here Rees and coworkers have been able to obtain structures of nitrogenase in the absence of HCA at atomic resolution using cryo-electron microscopy. These structural data have been long awaited. Additionally, authors have obtained a structure of a complex between HCA-deficient nitrogenase and a previously uncharacterized protein NafT.

Significance

Will the work be of significance to the field and related fields? How does it compare to the established literature? If the work is not original, please provide relevant references.

Yes, the work will be significant to the field and yes, it is original. Multiple novel structures of nitrogenase are presented. Nitrogenase is an extremely important enzyme as it has the potential of providing a sustainable alternative to the energy-consuming Haber-Bosch process of nitrogen fixation. The scientific community will be very interested in this work.

Data and Methodology

Does the work support the conclusions and claims, or is additional evidence needed?

Yes, the work does support the conclusions and claims – additional evidence I suggested is under the suggested improvements section.

Is the methodology sound? Does the work meet the expected standards in your field?

Yes, the methodology is sound. CryoEM is a quickly evolving field and Rees and coworkers have utilized the most recent tools to solving single particle EM.

Is there enough detail provided in the methods for the work to be reproduced?

Yes, there is enough detail provided to reproduce the results.

Validity

Are there any flaws in the data analysis, interpretation and conclusions? - Do these prohibit publication or require revision?

I do not believe there are flaws in the data analysis, interpretation, and conclusions.

Clarity and context/ References

The text is clear and the references the literature appropriately. There are two sections I would like a few more references and they are noted in the “suggested improvements” section.

Suggested improvements

1. P 5 line 112 "Intentionally blurred maps" are mentioned in the text and not explained further in methods

RESPONSE: We have altered the main text from 'intentionally blurred maps' to 'smoothed maps' for clarity. We have also added a section within the methods (page 17, lines 407-411) to describe how the smoothed maps were generated.

2. Page 6 Line 119 has a typo: "The additional the significance"

RESPONSE: Thank you. We have corrected this to "the additional significance."

3. Page 9 Line 204 and Extended figure 3 and 8. In some structures shown in Ext Fig 8, it looks like there is density for a metal in the histidine coordination site and there is an orange sphere shown, but authors do not seem to comment on the orange sphere. Has a metal ion been refined in any of the structures? Is the "mononuclear metal site" in Ext Fig 3, the same thing as the 'histidine coordination site." Clarity on these points would be helpful.

RESPONSE: Thank you for pointing this out – we have corrected the color scheme of Extended Data Figure 8 to reflect the unknown "UNX" atom that was refined into the 3DVA models. To further clarify the distinction between this coordination site and the mononuclear metal site, we have added text to the Extended Data Figure 3 legend that identifies this latter site as distinct from that shown in Extended Data Figure 8.

4. Page 9 Lines 190-191 I don't think that it is clear that the experimental cryoEM density was good enough to validate that NafT is the protein bound to the MoFe protein. If true, it would be good to say that you were able to fit the NafT sequence into the map and validate the AlphaFold prediction. It would be good to show density for sidechains of NafT and the sequence of NafT as an extended figure.

RESPONSE: We have modified the main text to indicate that the density matches the AlphaFold prediction (page 9, lines 195-196), and have further added a panel to Figure 5 (page 24, Figure 5, new panel b). This panel emphasizes the fit of NafT to the cryoEM density and the validation of the AlphaFold model by residue Q-scores (model fit to density). We also show visual representations of select residues fit to the cryoEM density.

5. Page 9 Lines 205-206 "While the function of the NafT protein has yet to be elucidated, it is clear that it recognizes homocitrate-deficient forms reminiscent of the MoFeAlkaline-inactivated state." I am assuming authors mean that the presence of NafT bound in the EM structure is what makes the recognition clear, but authors might want to rephrase "... it is clear from the structural data presented here..."

RESPONSE: Thank you. We have clarified this text to read "this structure demonstrates..." (page 9, lines 211-212).

Reviewer #2 (Remarks to the Author):

I was asked to review the manuscript because of my expertise in cryo-electron microscopy and not because I am familiar with nitrogenase. The work is exemplary in terms of cryoEM analysis. The resolution is better than average and the models do not show any obvious errors. Also, the fact that CET was done to investigate AWI gives the work a certain charm.

RESPONSE: Thank you.

There are only two things I would like to see. First, AWI is a normal problem in the field, and not just specific to nitrogenase. Here I would like to see a bit more discussion. This is something almost every cryoEM study has to take into account.

RESPONSE: We have added more discussion of the air water interface in the main text from pages 3-4, lines 59 – 68, namely highlighting the effects of the AWI; adding relevant references including Glaeser 2018 and Glaeser and Han 2017; and emphasizing the importance of cryoET in grid optimization.

Personally, I find the small protein NafT the most intriguing. Is this an assembly intermediate or a way to eliminate inactive complexes - dead end assembly? This should be speculated a little more. Especially in the context of other assembly intermediates of other large machines like complex I or PSII. This would place the nitrogenase here in a larger context of the biogenesis of molecular machines.

RESPONSE: In the final paragraph we have added more discussion of the possible role of NafT in a physiological context with references to photosystem II as an example (page 11, lines 250 – 260).

Reviewer #3 (Remarks to the Author):

I find this paper to be interesting and scholarly. It is clearly written and to my view the experiments are carefully carried out and reported. The mechanism of nitrogenase has been challenging to obtain given all the E-states and difficulties synchronizing the transitions between, but progress has been made by a variety of groups, including of course the Rees laboratory and associates. Thanks to Rees and others we have had a resting state structure of the MoFeCo cluster and the P-cluster and the surrounding protein for ~3 decades now. Some of the popular mechanistic models basically are tuned around a rather static structure will little changing through the states (eg the Hoffman et al Janus/hydrige elimination model) But now cryoEM is showing interesting changes (this manuscript, and the recent Tezcan et al Science paper (ref 11 here). The changes of homocitrate binding observed here, along with an number of parallel observations, are quite interesting. Though this is not a final story as of yet, it points to a more dynamics nitrogenase enzyme structure which should lead to a better idea of how this crucial enzyme actually works. So I am very supportive of publication.

RESPONSE: Thank you.

Reviewer #4 (Remarks to the Author):

Warmack et al. aim to describe the role of homocitrate molecule in the catalytic cycle of reducing dinitrogen.

Figure 2 they indicate that in panel e they indicate a drop in Fe from 33 to 26 moles per mole of protein. However, the mean in the graph does not reflect these values. From the graph displayed the values appear to be 30.5 and maybe 28 in the alkaline-inactivated version. The abstract and figure 2. Indicate a molar ratio of 7 Fe and 1 Mo per mole of protein in the dinitrogenase and a 8Fe in the P-cluster thus a total of 15. They measure ~30. How do they reconcile the discrepancy? Appears they are using the "dimer" complex molecular weight. This is ok but the axis needs to be updated for accuracy.

RESPONSE: Thank you for pointing these out – we have updated the legend of Figure 2 and the main text to reflect the correct averages of moles Fe represented by panel e (main text: page 5, line 109). To clarify what we mean by the molar ratios of metal to protein, we have added moles metal per "tetrameric" protein within the legend of Figure 2.

Similarly in (2f) the values in the text don't match the graph. The text indicates mean of 2 and 1.2 but interpolation from the graph yields 1.8 and 1.4 are closer to what is graphed.

RESPONSE: We have updated the legend of Figure 2 to reflect the correct averages of moles Mo represented by panel f.

For both 2f and 2e they fail to provide any statistical test. Based on the variance in Fe I would suggest that the decrease fails to be significant. They would likely need more replicates to have the power to detect the level of Fe decrease shown. Similar comment for Mo. There was no discussion what the potential decrease meant.

RESPONSE: To demonstrate that the modest decreases seen in Fe and Mo between the $\text{MoFe}^{\text{Alkaline}}$ and $\text{MoFe}^{\text{Alkaline-inactivated}}$ states are not significant we have conducted a paired t-test and included the calculated P-values in Figure 2. We have added that this analysis suggests the differences in metal contents are statistically insignificant within the main text (page 5, line 109).

The legend of Figure 2 is missing a description of what color each ball represents.

RESPONSE: We have added a description of the color coding within the legend of Figure 2 and Figure 4.

For the ICP-MS methods they indicate a volume of sample used but not the concentration. Thus it is not possible to deduce the standard curve concentration range used. The information on the standard curve used should be included. Were internal standards used and which ones.

RESPONSE: The ICP-MS methods section has been modified to include the concentration range of samples and standards utilized (page 14, lines 323-333).

More detailed methods for the ICP-MS are required to reproduce this study. What flow rate of oxygen was used? Also why did they not conduct a mass shift assay if they are using oxygen. Particularly for Mo as it readily forms the double oxide. With a relatively pure sample it seems that using He KED would be appropriate too.

RESPONSE: Thank you for highlighting this. The ICP-MS section in the Methods has been modified to include the flow rate of oxygen (0.4 mL/min), and to clarify the use of the

mass shift for the oxygen mode, as well as parallel use of helium mode to analyze the samples (page 14, lines 323-333).